# Language Experience Is Associated with Infants’ Visual Attention to Speakers

**DOI:** 10.3390/brainsci10080550

**Published:** 2020-08-13

**Authors:** Natsuki Atagi, Scott P. Johnson

**Affiliations:** 1Department of Child and Adolescent Studies, California State University, Fullerton, CA 92831, USA; 2Department of Psychology, University of California, Los Angeles, CA 90095, USA; scott.johnson@ucla.edu

**Keywords:** infancy, bilingualism, visual social attention

## Abstract

Early social-linguistic experience influences infants’ attention to faces but little is known about how infants attend to the faces of speakers engaging in conversation. Here, we examine how monolingual and bilingual infants attended to speakers during a conversation, and we tested for the possibility that infants’ visual attention may be modulated by familiarity with the language being spoken. We recorded eye movements in monolingual and bilingual 15-to-24-month-olds as they watched video clips of speakers using infant-directed speech while conversing in a familiar or unfamiliar language, with each other and to the infant. Overall, findings suggest that bilingual infants visually shift attention to a speaker prior to speech onset more when an unfamiliar, rather than a familiar, language is being spoken. However, this same effect was not found for monolingual infants. Thus, infants’ familiarity with the language being spoken, and perhaps their language experiences, may modulate infants’ visual attention to speakers.

## 1. Introduction

Bilingual experience modulates preverbal infants’ visual attention to talking faces (see, e.g., References [1,2,3,4,5], cf. References [6,7]). For instance, 4-to-12-month-old bilinguals have been found to look longer at the mouths of talking faces than do their monolingual peers [3]. Moreover, 8-month-old bilinguals can visually discriminate changes in the language being spoken when watching silent videos of talking faces, even though monolingual infants of the same age are unable to detect such changes [4,5]. In addition, bilingual infants detect such changes in languages with which they are familiar [5], as well as languages with which they are unfamiliar [4]. However, it is largely unknown whether and how attention to familiar versus unfamiliar linguistic stimuli may vary in bilingual infants as they develop productive language skills (cf. References [2,6]). Here, we examined how language experiences shape the ways in which infants attend to speakers during a period of rapid language growth between 15 and 24 months of age (see, e.g., References [8,9]). Specifically, we examined how infants’ language experiences and familiarity with the language being spoken may modulate their visual attention to speakers engaging in conversation.

Typically, such studies of bilingual infants’ visual attention to faces involve video clips of a single speaker talking as infants’ visual attention to the speaker’s face is examined. Across trials, infants may see multiple speakers, but in each trial, infants usually only view one speaker at a time speaking in one language at a time. For instance, Weikum and colleagues [5] showed infants videos of three bilingual French-English speakers; each speaker was featured individually in each trial, and each speaker spoke in French in one trial and English in another trial. Although studies employing these methods have generally demonstrated that infants prefer to look at the faces of familiar language speakers (e.g., Kinzler et al. [10]) and that bilingual infants show protracted attention to the mouth region of a face (e.g., References [2,3], cf. References [6,7]), infants’ real-life language input is more complex and frequently involves engaging with and attending to multiple speakers conversing, (see, e.g., References [11,12]).

Indeed, studies of infants’ perception of conversations suggest that infants rapidly develop an understanding of conversational turn-taking (see, e.g., References [13,14,15,16,17,18]). Infants as young as 8 to 21 weeks of age engage in turn-taking behaviors in proto-conversations [17], and this skill changes dynamically with age and productive language development over the first 18 months of postnatal life [18]. Additionally, eye-tracking evidence suggests that infants are able to make anticipatory gaze shifts to predict turn-taking structure in a conversation by 2 years of age [14] (cf. Keitel and Daum [15]) and that this ability improves with age [14,15,16]. Moreover, the speech cues that drive this early ability to predict conversational turn-taking seem to change with development: anticipatory gaze shifts to predict turn-taking stem from attention to prosodic cues early in development (e.g., in 1- and 2-year-olds [14,19] and in 3- but not 1-year-olds [15]) and may shift to rely more on lexicosyntactic/semantic cues over the course of development [19]. However, the effects of infants’ language experiences—particularly, their experience with more than one language—on their ability to anticipate conversational turn-taking are not well understood. Thus, to examine how infants’ language experiences shape the ways in which infants dynamically attend to speakers communicating with one another, the present study presented monolingual and bilingual infants with two speakers engaging in conversation in two languages. One language was familiar, and infants had access to both prosodic and lexicosyntactic/semantic cues in the speech. The other language was unfamiliar, thus infants only had access to prosodic cues in the speech.

Although studies of infants’ understanding of conversational turn-taking have not explicitly examined the impact of bilingual experiences, evidence from other literature suggests that infants’ early language experiences may shape their communicative development. Indeed, young bilingual children are sensitive to the communicative context of their environment (see, e.g., References [20,21,22]), and bilingual experiences may increase children’s attention to nonlinguistic communicative cues (see, e.g., References [23,24,25,26,27,28]). For example, bilingual children as young as 2 years of age have been found to repair communication breakdowns by switching languages to match the language being used by their communication partner [20]. Such findings suggest that bilingual infants are able to not only discriminate between languages but also dynamically modify their language use within a conversation to fit the needs of their interlocutor. Additionally, bilingual children attend to nonlinguistic cues more than monolingual children (gestures [25], eye gaze [28], and other referential cues (see, e.g., References [26,27])) to aid comprehension during communication. Moreover, even minimal exposure to a bilingual environment may help infants and young children develop more advanced perspective-taking skills [23,24]. Thus, bilingual experience may heighten sensitivity to both linguistic and nonlinguistic cues during communication.

The attention processing account of bilingualism [29] provides one possible explanation for why bilingual infants may attend differently to talking faces and communicative cues. According to this account, bilingual experience shapes attentional processing in infants because bilingual environments add complexity to an infant’s world. This is because infants are exposed to two sets of acoustic and statistical regularities—such as phonemes, grammars, and prosodies—as well as sociolinguistic features—such as speaker identities, facial characteristics, and pragmatics. Such complexity in bilingual environments is thought to provide infants with opportunities to compare and contrast the two sets of information, particularly attending to the differences. Thus, infants with exposure to bilingual environments may attend to multiple cues to help them process, organize, and represent the two languages. This theoretical account would suggest that bilingualism may modulate the way in which children attend to and take in information during communicative events, perhaps conferring an advantage in infants’ ability to process conversational structure.

### The Present Study

The present study examined visual attention in monolingual and bilingual 15-to-24-month-olds toward speakers engaging in conversation with the goal of elucidating how language experiences and familiarity with the language being spoken modulate visual social attention to individuals engaging in conversation. Infants participated in a free-viewing task in which they watched a series of six video clips with two native Armenian-English bilinguals conversing with each other; in each video clip, the speakers conversed in either English (a familiar language, i.e., a language in which infants could understand both prosodic and lexicosyntactic/semantic cues) or Armenian (an unfamiliar language, i.e., a language in which infants could understand only prosodic cues). Parents completed the MacArthur–Bates Communicative Development Inventory: Words and Sentences (MCDI) [8,9], which was used to measure infants’ English language development, as well as the communication scale of the Vineland Adaptive Behavior Scales (Vineland-II) [30], which was used to assess infants’ broader communicative development.

In line with the attentional processing account of bilingualism [29] and reported differences between monolingual and bilingual infants’ visual attention to speakers’ faces (see, e.g., References [2,3,4,5], cf. References [6,7]), we hypothesized that infants’ visual attention to a conversational structure would also vary by bilingual experience. Specifically, given previous findings that even minimal bilingual exposure may heighten infants’ and young children’s attention to and use of nonlinguistic communicative cues (see, e.g., References [23,24,25,26,27,28]), we expected infants’ visual attention to speakers to vary as a function of the infants’ language experiences. Specifically, we predicted that bilingual infants would anticipate conversational turn-taking more than monolinguals by looking earlier toward the next speaker in turn during a conversation. We operationalized this as the proportion of anticipatory gaze shifts prior to speech onset (latency < 100 ms; see, e.g., Leigh and Zee [31]), for each speaker as the conversations ensued. 

## 2. Materials and Methods

### 2.1. Participants

#### 2.1.1. Participants’ Ages and Recruiting

Participants were 15-to-24-month-old monolinguals (*n* = 34, 23 males; M_age_ = 19.48 months, SD = 2.78, range = 15.62–23.87) and bilinguals (*n* = 43, 28 males; M_age_ = 19.07 months, SD = 3.16, range = 14.79–23.93) from a large city in the United States. Ten additional infants participated but were excluded from the analyses due to poor data quality from excessive movement or extreme fussiness such that an eye-tracking session did not begin. All infants were recruited from a university’s child participant database. Infants were provided with a gift for participating.

#### 2.1.2. Participants’ Characteristics

Infants were categorized as either monolingual or bilingual based on parent-reported information about the amount of English exposure infants received on an average day. Parents completed three questionnaires to provide information about their infant’s language and communicative development: a language and demographic background questionnaire, the American English version of the MCDI [8,9], and the Vineland-II [30]. 

First, the language and demographic background questionnaire provided information about the languages spoken in the household, estimates of the percentage of exposure to each language, and parental education status. In line with other studies examining monolingual and bilingual children in linguistically diverse cities (see, e.g., References [7,32]), infants with more than 90% exposure to English were categorized as monolingual, and those with 20–80% exposure to English were categorized as bilingual. Infants were exposed to a variety of non-English languages (see Table 1 for the full demographic information). Critically, because of the way that the stimuli were constructed, children who were exposed to Armenian were not recruited for the study. Monolingual and bilingual infants’ primary caregivers did not statistically differ in their educational attainment (Mann–Whitney *U* = 637.500, *Z* = −0.269, *p* = 0.788), with 84.85% (28 out of 33) of monolingual and 80.00% (32 out of 40) of bilingual infants’ primary caregivers having a four-year college degree or higher; one monolingual infant’s primary caregiver and three bilingual infants’ primary caregivers did not report their educational attainment.

Second, the MCDI, which is a standardized checklist of children’s vocabulary, provided information about the number of words infants understood and produced. All infants understood more words than they produced, and monolingual and bilingual infants in this sample did not differ in the number of English words that they understood or produced (see Table 2). Lastly, the Vineland-II—a standardized assessment of children’s adaptive behaviors, in which parents rate how frequently their children engage in different behaviors (0 = never, 1 = sometimes or partially, 2 = usually)—provided a measurement of adaptive skills. Here, we focused on the skillset in the domain of communication (e.g., looks toward the parent or caregiver when hearing the parent’s or caregiver’s voice, waves goodbye when another person waves or the parent or caregiver tells him or her to wave). Bilingual infants scored higher in the communication domain of the Vineland-II than monolingual infants. See Table 2 for further information and statistics about the sample characteristics.

### 2.2. Apparatus

Stimuli were presented on a ViewSonic VX2268wm monitor with a 47.4 × 29.6 cm display, where stimuli filled the entire screen. Infants were seated approximately 60 cm from the display on their caregiver’s lap. Infants had a marker placed on their forehead and eye-movement data were collected via an EyeLink 1000 eye tracker (SR Research, Kanata, ON, Canada). Eye movements were recorded at 500 Hz with a spatial accuracy of approximately 0.5°–1° visual angle and down-sampled to 60 Hz to match the maximum refresh rate of the computer monitor. Experimental stimuli were generated with Experiment Builder (Version 1.10.1630), proprietary software of the EyeLink 1000 eye tracker (SR Research, Kanata, ON, Canada).

### 2.3. Procedure

Parents completed all questionnaires (described in Section 2.1.2) before infants were observed in the free-viewing task. Prior to eye-tracking, a five-point calibration scheme was used to calibrate each infant’s point-of-gaze (POG) and was repeated until the POG was, at maximum, within 1° of the center of the target. The eye-tracking session began only after the calibration criterion had been reached.

#### Free-Viewing Task

The free-viewing task was used to measure infants’ dynamic visual attention to speakers communicating with one another. Infants’ eye movements were recorded as they viewed six 20–25-second videos of two female English-Armenian bilingual speakers conversing with each other and to the infant in infant-directed speech to simulate a three-person conversation with the infant. Because infants were seated on their caregiver’s lap during this task, caregivers were asked to hold their infants on their lap, with the infant facing the monitor, and try to keep their infants as still as possible.

In all six videos that the infants watched, the speakers were seated on a couch in a child-friendly playroom (see Figure 1). The woman seated on the left was coded as speaker 1, and the woman on the right was coded as speaker 2. The dialogue in three of the videos was in Armenian, a language unfamiliar to all infants, and the dialogue in the other three videos was in English, a language to which all infants in the study had some amount of exposure. The women did not code-mix or -switch between English and Armenian during any of the videos; that is, in three of the videos, the women spoke only in English to each other, and in the other three videos, the women spoke only in Armenian to each other. The content of the dialogue in each video involved scenarios familiar to infants—morning routines, going to the zoo, and being hungry/eating—and the dialogue about these three scenarios were filmed in English and in Armenian. Thus, the English and Armenian videos contained dialogue that were translations of each other. In each video, there were eight speech onsets when the conversation switched from one speaker to the other, and across the three conversations in each language, there were a total of seven questions (range: 1–4 questions per conversation). The video order was pseudorandomized for each infant, such that no more than two consecutive videos were presented in the same language, and between each video, an attention-getter fixation stimulus appeared in the center of the screen to reorient the infants’ attention.

Each frame of every video was hand-traced for the two women’s faces and hand-coded for head turns (i.e., the direction in which the women’s faces were facing). The two women’s faces were the primary areas of interest (AOIs), whereas all other aspects of the scene were coded as a separate AOI. 

### 2.4. Analysis of the Eye-Tracking Data

Fixations directed at each of the three AOIs, namely, the two women’s faces and the background, were calculated using software written in MATLAB R2018b, version 9.5 (MathWorks, Natick, MA, USA). In line with a previous eye-tracking study of infants’ visual social attention (see, e.g., Tsang et al. [7]), a fixation was defined as lasting at least 300 ms (also see Gredebäck et al. [33] for a discussion about filtering fixations in infant eye-tracking research). Using this criterion, the latency to look at the two women’s faces was assessed.

The primary measure of interest in the free-viewing task was the latency to look at each speaker’s face (measured in milliseconds) with the onset of speech, which provided a measure of infants’ attention to each speaker during the conversation. If the latency to look at each speaker’s face was <100 ms before the speech onset, these gaze shifts were considered anticipations (see, e.g., Leigh and Zee [31]), presumably stemming from the infants’ processing of turn-taking during conversation (see, e.g., References [14,15,16,19]). If the latency to look at each speaker’s face with the onset of speech was >100 ms, these gaze shifts were considered to be reactions, i.e., the attention was given to the speaker due to speech onset. This yielded a binary value, i.e., whether the gaze shifts were anticipations or reactions.

## 3. Results

The present study aimed to understand how infants’ language experiences and familiarity with the language being spoken may modulate their visual social attention. We first report preliminary analyses of the infants’ language and communicative skills. Next, we report initial analyses of baseline looking measures, in which we describe data from the free-viewing task. Lastly, we report analyses examining group differences between monolingual and bilingual infants’ visual social attention to speakers conversing in a familiar versus unfamiliar language. In particular, we focus on the ways in which monolingual versus bilingual infants’ visual attention vary as a function of speech onset or speakers’ nonlinguistic communicative cues (i.e., head turns).

### 3.1. Preliminary Analyses

Infants’ language and communicative skills were assessed using parent-completed measures. Monolingual and bilingual infants did not significantly differ in the number of English words they were reported to understand (monolingual: M = 269.23, SD = 182.48; bilingual: M = 227.95, SD = 171.82) or produce (monolingual: M = 121.19, SD = 178.98; bilingual: M = 116.35, SD = 142.36; both *p*s > 0.30) on the MCDI (see Table 2). However, bilingual infants (M = 130.80, SD = 23.88) were scored significantly higher than monolingual infants (M = 115.69, SD = 29.99) in the domain of communication on the Vineland-II (*t*(71) = −2.399, *p* = 0.019, Cohen’s *D* = 0.566). Thus, although the size of the monolingual and bilingual infants’ English vocabulary did not differ, bilingual infants in this sample were reported to have more advanced communicative skills than their monolingual counterparts.

We also examined baseline looking during the free-viewing task. The percent of time infants spent attending to the screen (e.g., on-task attention) did not differ by the infant’s language background (*t*(75) = 0.053, *p* = 0.957, Cohen’s *D* = 0.012) nor vary by the infant’s age (*r* = −0.02, *p* = 0.889). On-task attention among infants also did not differ when Armenian (unfamiliar language) or English (familiar language) was spoken (*F*(1,74) = 0.984, *p* = 0.324, partial *η*^2^ = 0.013).

### 3.2. Comparing Monolingual and Bilingual Infants’ Visual Social Attention to Speakers

First, to compare monolingual and bilingual infants’ overall attention to the speakers’ faces, we conducted a 2 (language background: monolingual vs. bilingual) × 2 (language familiarity: familiar vs. unfamiliar) repeated measures ANCOVA covarying for the effects of age. No main effects of the infants’ language background (*F*(1,73) = 0.003, *p* = 0.960, partial *η*^2^ < 0.0001) or language familiarity (*F*(1,73) = 0.004, *p* = 0.950, partial *η*^2^ < 0.0001) were found, nor was a language background by language familiarity interaction found (*F*(1,73) = 0.767, *p* = 0.384, partial *η*^2^ = 0.01). When this same analysis was conducted by covarying for Vineland-II communication scores instead of age, similar result patterns were found.

We next examined whether the order in which languages were presented to the infants impacted the infants’ overall attention to faces. Although the video order was pseudorandomized for each infant, the first trial for half of the infants was in English (familiar language), and the first trial for the other half of infants was in Armenian (unfamiliar language). We expected overall attention to faces across the six videos to not differ between infants whose first trial was in English versus Armenian. A 2 (language background: monolingual vs. bilingual) × 2 (language familiarity: familiar vs. unfamiliar) × 2 (language of trial 1: English vs. Armenian) ANCOVA covarying for age revealed no significant main effects or interactions. However, a marginal interaction between language familiarity and the language of trial 1 was noted (*F*(1,72) = 3.577, *p* = 0.06, partial *η*^2^ = 0.05; all other main effects and interactions: *F*(1,72) = 0.001–1.395, *p* = 0.24–0.97, partial *η*^2^ = 0.00002–0.02): infants made more fixations to the face when watching familiar language (English) videos (M = 64.62%, SD = 12.46%) than unfamiliar language (Armenian) videos (M = 60.99%, SD = 12.16%) when their first trial was in English, whereas infants did not differ in their overall fixations to the face for familiar (M = 60.45%, SD = 18.74%) versus unfamiliar (M = 60.54%, SD = 14.24%) language videos when their first trial was in Armenian. However, when this same analysis was conducted covarying for Vineland-II communication scores instead of age, this marginal interaction disappeared (*F*(1,69) = 1.591, *p* = 0.21, partial *η*^2^ = 0.023); all other patterns of results remained similar to the original analysis covarying for age.

Next, we compared the latency to shift attention to the speaker at the onset of speech. We hypothesized that bilingual infants would be more attentive to turn-taking during communication than monolingual infants, and thus, produce more anticipatory gaze shifts (i.e., shifts in attention <100 ms before speech onset) to look at the next speaker. Moreover, given that the attentional processing account of bilingualism [29] posits that bilingualism may especially heighten infants’ attention to differences in the sociolinguistic environment and studies have found bilingualism to increase infants’ and young children’s attention to nonlinguistic communicative cues [23,24,25,26,27,28], we expected the tendency for bilingual infants to engage in anticipatory looking would be particularly strong when the speakers were using an unfamiliar language and infants only had access to prosodic cues. Because the dependent variable was a binary response variable (i.e., for any given instance of speech onset in each video clip, infants either made or did not make an anticipatory gaze shift), we conducted a two-proportions *Z*-test to compare whether the proportion of anticipatory gaze shifts were higher between monolingual and bilingual infants when the speaker was speaking an unfamiliar language (Armenian) than a familiar language (English). Monolingual and bilingual infants did not differ in their proportion of anticipatory gaze shifts for either familiar (monolingual = 55.88% vs. bilingual = 46.51%; *X*^2^(1) = 0.345, *p* = 0.557, 95% CI = (−0.34, 0.15)) or unfamiliar languages (monolingual = 82.35% vs. bilingual = 83.72%; *X*^2^(1) = 0, *p* > 0.99, 95% CI = (−0.169, 0.196)). However, bilingual infants engaged in significantly more anticipatory looks to the speaker when the language of the video clip was an unfamiliar language than a familiar language (*X*^2^(1) = 11.518, *p* < 0.001, 95% CI = (0.163, 0.581); *α* corrected for multiple comparisons = 0.05/2 = 0.025). The same effect was not observed for monolingual infants (*X*^2^(1) = 4.409, *p* = 0.035, 95% CI = (−0.505, −0.025); *α* corrected for multiple comparisons = 0.05/2 = 0.025; see Figure 2). It should also be noted that *t*-tests confirmed that these results were not simply reflecting differences between monolingual and bilingual infants’ Vineland-II communication scores since the Vineland-II communication scores were not significantly different between infants who displayed anticipatory versus reactionary gaze shifts during the free-viewing task (*t*(71) = 0.867, *p* = 0.389, Cohen’s *D* = 0.203). Thus, to summarize, although the proportion of anticipatory gaze shifts did not differ between bilingual and monolingual infants, bilingual infants made more anticipatory gaze shifts for unfamiliar than familiar languages. Monolingual infants’ proportion of anticipatory gaze shifts did not differ by language familiarity.

## 4. Discussion

The goal of the present study was to examine how infants’ language backgrounds and familiarity with the language being spoken may modulate infants’ visual social attention to speakers engaging in a conversation. Although our findings revealed that monolingual and bilingual infants did not differ in their overall visual social attention to speakers conversing in a familiar (English) versus unfamiliar (Armenian) language, monolingual and bilingual infants were found to differ in their patterns of engagement in anticipatory looking. Specifically, bilingual infants engaged in more anticipatory looks to speakers conversing in a familiar language—that is, a language in which they had access to prosodic and lexicosyntactic/semantic cues (see, e.g., References [14,15,16,19])—than an unfamiliar language—that is, a language in which they only had access to prosodic cues (see, e.g., References [14,15,16,19]); in contrast, monolingual infants’ proportion of anticipatory looks did not vary by their familiarity with the language being spoken.

One interpretation of these different patterns of anticipatory looks among monolingual and bilingual infants is that bilingual infants may have attended to the differences between the cues available in the familiar versus unfamiliar languages more than monolingual infants. This interpretation would lend partial support to the attentional processing account of bilingualism [29] and suggest that bilingual experiences may change infants’ attention to communicative cues. In particular, our findings would lend support to the idea that bilingualism may change the way in which infants attend to communicative cues during familiar communicative events, particularly when communicative events in a familiar language are juxtaposed with those in an unfamiliar language. Indeed, the attentional processing account of bilingualism posits that bilingualism may especially tune infants’ attention to differences in the sociolinguistic environment. Thus, bilingual infants anticipated turn-taking at different rates when a familiar versus unfamiliar language was being spoken, whereas monolingual infants did not differ in their anticipation of turn-taking for familiar versus unfamiliar languages. However, our findings only provide evidence of possible differences between the prediction mechanism underlying bilingual infants’ anticipation of conversational turn-taking in familiar versus unfamiliar languages and do not provide evidence of a difference in an overall attentional processing mechanism.

Another interpretation of the present findings may be that monolingual infants were more advantaged in anticipating turn-taking in the familiar language than bilingual infants. Although monolingual and bilingual infants did not statistically differ from one another in anticipating turn-taking for familiar or unfamiliar languages, the anticipation of turn-taking statistically differed for familiar versus unfamiliar languages among bilinguals but not monolinguals. This difference in patterns of anticipatory looking between monolinguals and bilinguals seemed to stem from the numerical difference between monolinguals’ and bilinguals’ anticipation of turn-taking for familiar languages: monolingual infants engaged in numerically more anticipatory gaze shifts when speakers were conversing in a familiar language than did bilingual infants. However, both monolingual and bilingual infants’ proportion of anticipatory gaze shifts while watching familiar language conversations was approximately 50%, suggesting that these gaze shifts may have been “anticipatory” by chance, rather than being systematically anticipatory. This is in line with Keitel and Daum’s [15] finding that 1-year-olds make many gaze shifts between speakers over the course of a conversation, rather than simply around speech onsets.

Yet another interpretation of these findings is that infants’ familiarity with the language being spoken may be more relevant for anticipating turn-taking than infants’ monolingual versus bilingual experience. The gaze of infants engaged in anticipatory gazing shifted more for the unfamiliar than familiar language (statistically more for bilinguals and numerically more for monolinguals), suggesting that the communicative cues available to infants during these conversations may impact infants’ ability to predict turn-taking (see, e.g., References [14,15,16,19]). Although infants had both prosodic and lexicosyntactic/semantic cues available to them when watching conversations in English, a language familiar to all infants in the study, infants only had prosodic cues available to them and lacked lexicosyntactic/semantic cues when watching conversations in Armenian, a language unfamiliar to all infants in the study. These findings corroborate findings from previous studies showing that infants rely more on prosodic cues early on to predict conversational turn-taking and gradually shift to rely more on lexicosyntactic/semantic cues over the course of development (see, e.g., References [14,15,16,19]).

Our findings also generally support and extend previously reported differences between monolingual and bilingual infants’ visual attention to speakers’ faces (see, e.g., References [1,2,3,4,5], cf. References [6,7]). Previous studies have focused largely on preverbal monolinguals’ and bilinguals’ attention to the mouth (see, e.g., References [1,3,7,34]) because the mouth provides redundant audiovisual speech cues that facilitate language learning (see, e.g., References [35,36,37]) and social-communicative development (see, e.g., References [2,34]). The present findings extend these previous findings by demonstrating that monolingual and bilingual infants’ visual attention to speakers continues to differ even after they begin saying their first words and further develop their productive language skills. However, our findings point to the possibility that the manner in which monolingual and bilingual infants’ visual attention differs may change with language development: infants may initially show differences in mouth-looking when they are preverbal, but may shift their attention to larger communicative cues once they begin talking. A longitudinal study following preverbal infants through toddlerhood would be necessary to examine such a possibility of whether bilingual experiences and language development may shape the developmental trajectories of infants’ visual social attention. This would also provide more insight into the specific ways in which monolingual and bilingual infants’ visual attention to conversational turn-taking may differ over the course of their development and whether language experience shapes the kinds of linguistic cues that infants use to predict turn-taking (see, e.g., References [14,15,16]). Additionally, including an examination of the location on the face (e.g., eyes, mouth) that infants attend to (see, e.g., References [1,2,3,6,7,34]) may elucidate how infants use different visual cues to understand communicative events.

We also found that bilingual infants scored significantly higher than monolingual infants in the domain of communication on the Vineland-II. In other words, bilingual infants were reported as engaging in communicative behaviors (e.g., looks toward the parent or caregiver when hearing the parent’s or caregiver’s voice) more frequently than monolingual infants. Indeed, laboratory-based studies have found that infants and young children from bilingual environments show advanced perspective-taking skills [23,24] and attend more to nonlinguistic communicative cues (see, e.g., References [25,26,27,28]) than peers from monolingual environments. Our finding regarding bilingual infants’ communication skills, as measured by the Vineland-II, is not only in line with findings from such previous laboratory-based studies, but also suggests that these enhanced communicative skills are observable in the everyday lives of bilingual infants.

A couple of notes should be made about our findings. First, although we discuss our findings in terms of attention to familiar versus unfamiliar languages, it should be noted that we only tested for infants’ attention to English, which was familiar to all infants, versus Armenian, which was unfamiliar to all infants. It is possible that testing the attention of these English-exposed infants to other unfamiliar languages (e.g., Finnish) would have yielded different results. Likewise, testing bilingual infants’ attention to their non-English language, which would have been another familiar language, may have yielded different results as well. Second, monolingual and bilingual infants did not differ in their baseline looking and overall attention to faces during the free-viewing task, but we do not know whether monolingual and bilingual infants differed in their baseline levels of attentional shifts. Thus, it is possible, though unlikely given previous literature (see, e.g., References [14,15,16,19]), that infants’ anticipatory and reactionary gaze shifts were simply spontaneous shifts in attention. Finally, it is also possible that there were individual differences in visual acuity in the infants we observed. In general, an infant’s visual acuity is well-developed by about 6 months of age (see, e.g., References [38,39,40,41]), as well as the ability to visually scan the environment and track moving objects and people. Although it seems unlikely that poor acuity would have had a prominent influence on our results, otherwise we would not have obtained interpretable findings, measuring and equating visual acuity across conditions would strengthen the designs of future studies.

## 5. Conclusions

Overall, the present findings support the attentional processing account of bilingualism [29] and provide evidence that monolingual and bilingual infants’ visual social attention may differ, even as they develop productive language skills. Moreover, the present study demonstrates that bilingual infants may attend to communicative cues differently than their monolingual peers, particularly during challenging communicative events. Early bilingual experience may thus shape the ways in which infants and young children attend to cues in their sociolinguistic environments.

## Figures and Tables

**Figure 1 brainsci-10-00550-f001:**
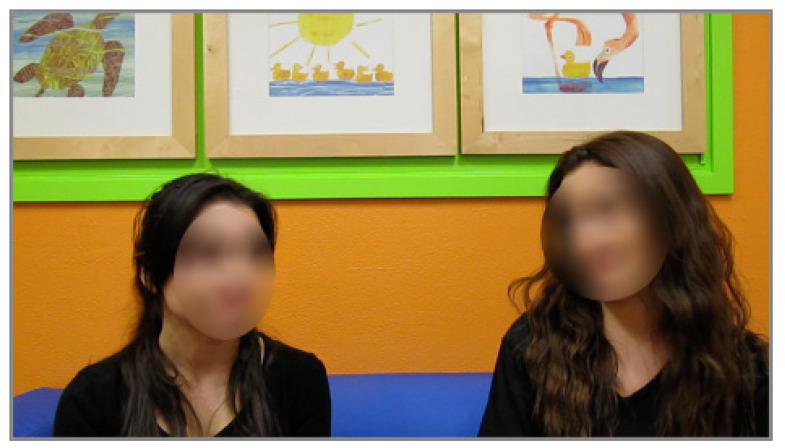
An example frame from a video used in the free-viewing task. In all six videos, the woman on the left (coded as speaker 1) was always seated on the left-hand side of the screen, and the woman on the right (coded as speaker 2) was always seated on the right-hand side of the screen. It should be noted that the faces are only blurred here to protect the privacy of the actors; the faces were not blurred in the actual stimuli presented to the infants.

**Figure 2 brainsci-10-00550-f002:**
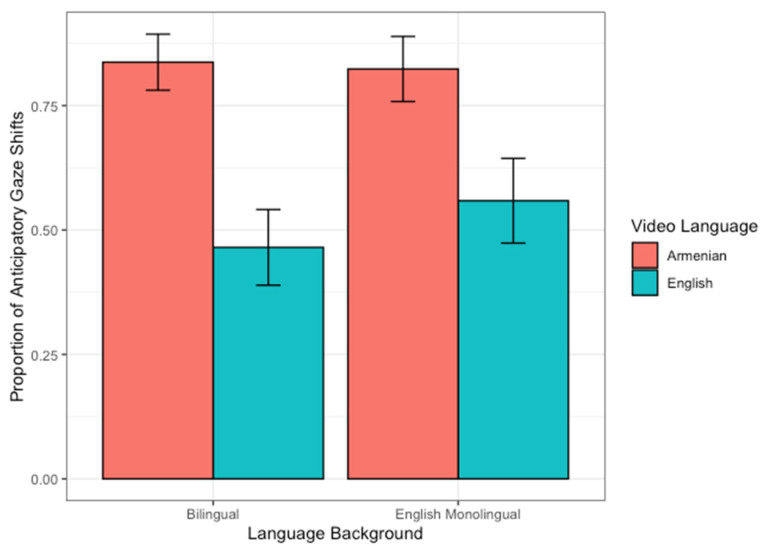
The proportion of anticipatory gaze shifts made by bilingual versus monolingual infants when viewing videos in which speakers spoke in an unfamiliar (Armenian) versus familiar (English) language. Error bars represent 95% confidence intervals.

**Table 1 brainsci-10-00550-t001:** Demographic information by language background.

Demographic Characteristics	Monolingual (%)	Bilingual (%)
Race/Ethnicity		
Asian	15	30
Black	0	2
Hispanic	3	28
Mixed	12	9
White	67	26
None reported	3	5
Non-English Languages		
Cantonese	3	5
Chinese	0	2
Danish	3	0
Farsi/Persian	0	7
French	0	7
Hebrew	0	2
Japanese	0	2
Korean	0	5
Mandarin	0	7
Nepali	0	2
Spanish	18	51
Swedish	0	2
Tagalog	3	2
Taiwanese	3	0
Urdu	0	2
None	71	0

**Table 2 brainsci-10-00550-t002:** Sample characteristics by language background.

Sample Characteristics	Monolingual (*n* = 34, 23 Male) M (SD)	Bilingual (*n* = 43, 28 Male) M (SD)	*t*	*df*	*p*
Age (in months)	19.48 (2.78)	19.07 (3.16)	0.585	75	0.560
% English exposure	99.26 (1.50)	55.28 (19.20)	13.309	75	<0.0001
English MCDI					
Understands	269.23 (182.48)	227.95 (171.82)	0.971	68	0.335
Says	121.19 (178.98)	116.35 (142.36)	0.128	70	0.899
Vineland-II					
Communication ^1^	115.69 (29.99)	130.80 (23.88)	−2.399	71	0.019

^1^ The Vineland-II is normalized by age such that a score of 100 is considered average.

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
