# Peer review of "Language Experience Is Associated with Infants’ Visual Attention to Speakers"

_brainsci, 2020, doi:10.3390/brainsci10080550_

Round 1

Reviewer 1 Report

Please refer to the attached Review.pdf for comments.

Reviewer 2 Report

This manuscript presents a study addressing the question as to whether there are differences between bilingual and monolingual infants in terms of their visual attention to speakers of a familiar or foreign language. There are a number of changes the authors could make to strengthen their paper.

First, it is not clear why the authors did the study in the first place, from a theoretical perspective. What exactly is it about the exposure to one or two languages that would lead to differences in how children attend to speakers in interactions based on familiarity/unfamiliarity of the language?

Second, because the theoretical rationale is not clear, some of the methodological choices are unclear. Particularly, why place two people next to each other speaking in two different languages? Why did they recruit bilinguals with a variety of backgrounds (I assume that is because it is simply THAT the children heard two languages that mattered, not which languages they heard, right?). But also, why analyze the results, covarying for age (why would age matter)?

Third, the authors could discuss whether they think they have an unusual sample. The bilingual children had equivalent ENGLISH vocabulary scores in both comprehension and production, even though they averaged about 55% of their exposure to English. Since many studies have shown that bilinguals tend to score lower in any one language than monolinguals, why do the bilinguals in this study seem so different? Could the results from the Vineland be any indication?

Finally, the interpretation of the results is not very convincing. While it was technically the case that the monolinguals did not show a statistically significant difference on anticipatory gazes between Armenian and English, that does rest on the need to correct for multiple statistics, and even then, the p is close to significant. The perceptual effect of Figure 2 is that there is no difference between bilinguals and monolinguals but a whopping difference between Armenian and English, with all the children doing more anticipatory looks to Armenian. That could be the more interesting result to frame this paper : i.e., why would children look more to an unfamiliar language, with little effect of bilingual/monolingual experience?

Smaller points (for some reason, this interface has replaced all of my "pages" with the numbers 1-10):

  1. 1, line 3 of text : longer than monolinguals or longer than silent speakers?
  2. 1, line 8  of text : I’m not terribly convinced that a study needs to be done simply because there are few studies on a topic. There are remarkably few studies testing the correlation between my body temperature and the amount of sunlight the North Pole gets. And I hope that situation remains the same.
  3. 2, line 3 : why would language experience impact (or not?) the visual attention?
  4. 2, second paragraph, line 8 : attend more than monolinguals or attend more to nonverbal cues than to verbal cues?
  5. 2, third paragraph, line 5 : the wording here is a touch awkward, since technically phonemes and grammars are explanatory constructs so this is not what children are actually exposed to…
  6. 2, fourth paragraph, toward end : why measure only English language development? Why not the other language, too?
  7. 2, fourth paragraph, toward end : why measure broader communicative development? How is that ability related to visual attention?
  8. 6, section 3.2, third line : why covary age and not, say, the Vineland (which differed on the two groups)?
  9. 6, last paragraph, second sentence : why did the authors hypothesize that the bilinguals would be more attentive to turn-taking? And how does their dependent variable relate to sensitivity to turn-taking? Why would it differ by familiar/unfamiliar language?
  10. 7, Figure 2 : what do error bars show?

Reviewer 3 Report

Please see comments in attachment

Round 2

Reviewer 1 Report

Review round 2

Language Experience is Associated with Infants’ Visual Attention to Speakers

Natsuki Atagi and Scott P. Johnson

Overall:

The manuscript has greatly improved. I have got only two minor requests for added information.

Specifically:

Regarding

  1. Did you test for visual acuity of the infants? Can you make sure that infants actually saw the stimuli and were able to interpret the non-verbal signals?

Your answer:

We did not test for infants' visual acuity, nor their understanding of nonverbal signals. However, infants' visual acuity is well-developed by about 6 months of age (e.g., Frantz, Ordy, & Udelf, 1962; Slater et al., 2010; Wang & Candy, 2010), and infants develop the ability to visually scan their environment and track moving objects over the course of the first 6 months of postnatal life (e.g., Johnson, Slemmer, & Amso, 2004). Moreover, in this study, no parents reported their infants as having vision problems. Additionally, given that infants' Vineland-II communication score--on average--was above the norm for their age, we have no reason to believe that infants were not typical for their age in their understanding of nonverbal cues. Regarding whether infants saw the stimuli, we have no reason to believe that infants couldn't see the stimuli. They were seated approximately 60 cm from the display, and we observed no unusual eye movements in the infants.

  • You are certainly correct that visual acuity develops rapidly in the first months. However, this is in general, not the individual. I am sorry to say that I had children of about 5 or six years in the lab, who were not able to discriminate quite big Stimuli on a screen about 60 to 70cm in front of them and neither them nor their parents had any clue about it. Unfortunately, this was not a N = 1 experience. I know you cannot go back and test children’s visual acuity. However, you should mention this in the discussion and please consider it in further research.

Regarding

  1. Please consider including a section “statistical analysis” containing information about the eye tracking data. Did you filter any of the eye movements? If yes, what were the criteria? How long did a fixation or dwell time have to be in order to count as a gaze shift?

Your answer:

We did not filter any of the eye movements, but our criteria for a fixation was at least 300 ms. Section 2.4. "Analysis of eye-tracking data" was added to include all of this information about the eye tracking data.

  • Please provide reasoning for using the 300ms criterion and a citation.

Reviewer 3 Report

The authors have addressed all my concerns. I just have two minor points: 

1) While they did adjust their interpretation of the data in terms of the direction of the effect within the main text, they haven't made these adjustments in the abstract and a similar change should be made there as well. 

2) At the end of the first paragraph of the discussion, the authors implicate the prediction mechanisms but do not make the appropriate citations, the prediction mechanism in turn-taking is not new and should be cited accordingly. 

Other than these two minor points, I have happy to recommend publication. 
